# Relationship between maltreatment and mental health in adolescents: A school-based study in Indonesia

**Nita Arisanti[1], Anisa Swediana[2], Deborah Karubaba[2], Anindita Noviandhari[3], Deni K. Sunjaya[1], Meita Dhamayanti[3]***

1 Department of Public Health, Faculty of Medicine, Universitas Padjadjaran, Bandung, Indonesia,
2 Profession Program, Faculty of Medicine, Universitas Padjadjaran, Bandung, Indonesia, 3 Department of Child Health, Faculty of Medicine, Universitas Padjadjaran, Bandung, Indonesia

* meita.dhamayanti@unpad.ac.id

**Data Availability Statement:** Our data cannot be shared publicly because of the high risk of disclosure associated with the confidential data sets. Data are available by requests to Research

## Abstract

Maltreatment affects emotional development in adolescents and inhibits social adjustment. This study aimed to analyze the relationship between maltreatment and mental health among adolescents. A cross-sectional study was conducted on adolescents in the first and second grades of middle school (12–14 years old) and high school (15–17 years old) in eight cities and municipalities in the province, selected through several stages of simple random sampling ($N = 1837$). The International Society for the Prevention of Child Abuse and Neglect (ISPCAN) Child Abuse Screening Tool for Children (ICAST-C) questionnaire for detecting maltreatment was translated, simplified, and validated by an expert based on a theoretical framework that involved pediatricians, public health, and medicolegal perspectives. The Strengths and Difficulties Questionnaire (SDQ) was used to assess emotional states. ICAST-C and SDQ scores were transformed to logit values using Rasch model analysis. Distribution frequency and linear regression were used for data analysis. The results indicated that 85.6% of adolescents aged 12–14 and 83% of those aged 15–17 experienced physical maltreatment, while 89.4% of the 12–14 age group and 82.9% of the 15–17 age group experienced psychological maltreatment. The emotional states of the two groups were 52.8% and 59.2%, respectively. There was a significant correlation between the experience of physical maltreatment and emotions among 12–14 ($r_1 = 0.148$ (0.190–0.257)) and 15–17 years old ($r_1 = 0.047$ (0.084–0.156)). There was a significant correlation between the experience of psychological maltreatment and emotions among 12–14 years old '($r_2 = 0.191$ (0.270–0.350)) and 15 to 17 years old ($r_2 = 0.097$ (0.167–0.252)). In conclusion, physical and psychological maltreatment were correlated with mental health states among adolescent students in West Java, Indonesia.

## Introduction

Adolescents are in a turbulent phase where they often argue with their parents or teachers and prefer to interact with their peers. This rebellious nature arises as a form of separation from

Ethics Committee Universitas Padjadjaran by email to kep@unpad.ac.id, for researchers who meet the criteria for access to confidential data.

**Funding:** The study was supported financially by an Academic Leadership Grant from Universitas Padjadjaran undernamed Meita Dhamayanti, and the APC was funded by Academic Leadership Grant from Universitas Padjadjaran 1 9 5 9 /UN6.3.1/ PT.00/2021 the funders had no role in study design, data collection and analysis, decision to publish, or preparation of the manuscript.

**Competing interests:** The authors have declared that no competing interests exist.

those who restrict adolescents from seeking their own identities. Unfortunately, this action is often not understood by parents and can lead to physical and psychological maltreatment. Maltreatment or abuse, is defined as the deliberate use of force against oneself, another person, or a group of communities, resulting in injury, death, psychological damage, retardation, or deprivation. As a result of this abuse, adolescents are increasingly motivated to fight their parents [1].

Maltreatment of children and adolescents has become a major global health problem comprising physical, psychological, sexual, and neglected health problems. It has been reported that one in four adults experienced maltreatment when they were adolescents. The prevalence of child maltreatment worldwide in recent years was 1 billion cases, and 50% were cumulatively found in Asia, Africa, and North America. Physical maltreatment was 226/1000, psychological or emotional abuse was 363/1000, physical neglect was 163/1000, and psychological or emotional neglect was 184/1000 [2, 3]. In Indonesia, 3 of 10 boys and 4 of 10 girls have experienced psychological maltreatment. Furthermore, 12 of 100 boys and 10 of 100 girls have experienced physical maltreatment [4]. According to the Indonesian Child Protection Commission (*KPAI*) survey, 91% of children experienced maltreatment in their homes, 87.6% in school, and 17.9% in the community. The impact of maltreatment may range from serious injury to mental health distortion and eventually death [5–7].

Emotional or mental health problems are common in adolescents, as they are in a transitional period, which makes them more vulnerable to maltreatment [8, 9]. One study reported that 450 million people worldwide have mental health problems, with the proportion of children and adolescents reaching 10–20% [10, 11]. A total of 104 out of 1000 children aged 4–15 years in the world experience mental health problems [12]. The prevalence of mental health problems among adolescents in India is 30% and 12.2% in Germany [13]. Various risk factors have been associated with mental health problems, including history of maltreatment or abuse in the family [14–17]. Various studies have stated that childhood maltreatment is a risk factor for mental health problems [18–20], but studies on the types of maltreatment and mental health problems are still limited.

In Indonesia, 45.5 million people (16.9% of the total population) are aged 10–19. West Java, Indonesia, has a significant population of 10–19-year-olds, totaling approximately 11.66 million people, which constitutes 27.1% of the region's total population [21, 22]. This is a reminder for the government to increase awareness of the risk of maltreatment in adolescents with other mental health problems [23]. This study hypothesized that physical and psychological maltreatment is correlated with mental health problems in adolescents. Hence, this study aimed to analyze the correlation between physical and psychological maltreatment and mental health problems among adolescents using the SDQ to assess the categories of mental health problems.

## Materials and methods

A cross-sectional study was performed on adolescents younger than 18 years and grouped by school level of the first and second grades of junior school (range = 12–14) and the first and second grades of high school (range = 15–17) in cities and municipalities of West Java Province, Indonesia. This study involved both government and private schools. We excluded students from the third grade of junior and high school because of their schedules for national exams. All protocols were performed in accordance with the national and international ethics guidelines for human participants. Ethical approval was obtained from the Health Research Ethics Committee of the Faculty of Medicine, Universitas Padjadjaran 244/UN6. C1.3.2/ KEPK/PN/2016.

The sampling method used was multistage random sampling. Consequently, four cities and four municipalities were selected for this study. Cities and municipalities were differentiated based on several characteristics. We performed two simple random samplings to select 16 junior and 16 high schools and then identified 48 junior and 48 high school classes. We invited 40 students from each class, representing grades 7 to 12.

The International Society for the Prevention of Child Abuse and Neglect (ISPCAN) Child Abuse Screening Tool for Children (ICAST-C) questionnaire was used to screen the children [21, 22]. An expert panel translated, selected, simplified, and tested the questionnaire for validation, followed by reliability tests [23]. The ICAST-C questionnaire consists of 19 and 17 questions on physical and psychological maltreatment, respectively. The responses for each question were interpreted as follows: Score 3 for the response of "Once or more in this year," score 2 for the response of "Not in the last one year but once in the lifetime," and score 1 for the response of "Never." The SDQ consists of 25 questions classified into 5 dimensions: prosocial, emotional, conduct, hyperactivity, and peer relationship problems. Each dimension consists of five questions. The response for each question was scored as 2 for "True," 1 for "Kind a True," and 0 for "Not True." The opposite value or response, score 2 for "Not True"; score 1 for "Kind a True"; and score 0 for "True," were applied for 5 specific questions and consisted of one question from the conduct problem dimension, two questions from prosocial dimension, and two questions from hyperactivity dimension [24].

## Data analysis

Data from the ICAST-C questionnaire were transformed by Rasch model analysis using a statistical software called Winstep version 3.73. Rasch model analysis assumes that behavior is determined by the difficulty of an item and the ability of a person [25]. The Rasch model analysis converts ordinal data into standardized ratio data called logit values (log odds units). Maltreatment is categorized if the logit value falls between x>M-1SD. If the logit value fell between x≤M-1SD, it was not categorized as maltreatment.

The score for each item in the SDQ was summed based on its subscale group and interpreted as normal, borderline, or abnormal, where the interpretation value for each subscale differed. The interpretation is described as follows: (1) prosocial dimension was categorized as *normal* with a value of 6–10, *borderline* with a value of 5, and *abnormal* with a value of 0–4; (2) emotional problem and hyperactivity dimension was categorized as *normal* if the value was 0–5, *borderline* if the value was 6, and *abnormal* if the value was 7–10; (3) conduct problem dimension was categorized as *normal* if the value was 0–3, *borderline* if the value was 4, and *abnormal* if the value was 5–10; (4) peer relationship problem, called *normal* if the value was 0–3, *borderline* if the value was 4–5, and *abnormal* when the value is 0–4. In addition to scores based on the subscales, this study used the sum of the difficulty dimension scores, consisting of emotional problems, conduct problems, hyperactivity, and peer relationship problems, which are referred to as *total problems* in this study. Then, the total problem value was interpreted as *normal* if it was 0–15, *borderline* if it was 16–19, and *abnormal* if it was 20–40. A higher total problem score indicated a higher level of mental health disorders, except for the prosocial dimensions; the higher the prosocial dimension score, the lower the level of mental health disorders. Mental health problems were defined as scores falling in the "abnormal" category.

The Rasch model analysis was used to convert the SDQ data into logit values. Differential Item Functioning (DIF) analysis was also used to assess the benefit of each item from both questionnaires to one of the groups so that it could be determined whether the question needed to be reviewed. Mental health problems were defined by the total score of each item in the questionnaire. Physical and psychological maltreatment and mental health problems in the

**Table 1. Reliability of the instruments.**

| Psychometric attribute | Instruments | |
|---|---|---|
| | ICAST | SDQ |
| Outfit mean square | | |
| Mean | 1.09 | 1.04 |
| Standard deviation | 0.44 | 0.17 |
| Reliability | 0.99 | 1 |
| Item separation | 11.83 | 19.05 |
| Cronbach's alpha | 0.8 | 0.7 |

form of logit values were further analyzed with a linear regression test using IBM SPSS Statistics 25. The correlation was considered positive if the coefficient of the relationship was r> 0.00. Correlation is considered weak if r = 0.1–0.3; moderate if r = 0.4–0.5; and strong if r> 0.5 with significant at p-value <0.05.

**Reliability and data fit.** The reliability of the questionnaire was measured using three indices: item reliability (> 0.67), item separation (> 3), and Cronbach's alpha (> 0.67). The reliability of the ICAST and SDQ demonstrated that all questionnaires were considered very good (Table 1).

## Results

A total of 1837 respondents were assessed by and completed the ICAST-C and SDQ, respectively (participant rate was 53%). Table 2 displays respondents' characteristics based on several categories. Most respondents were female, lived in rural areas, and had an intermediate level of parental education.

Table 3 shows that having experienced twisted ears, being forbidden from going out, and feeling unimportant were the most frequent types of maltreatment experienced by respondents.

**Table 2. Distribution of sociodemographic characteristics.**

| No. | Characteristics | Participants | |
|---|---|---|---|
| | | N | % |
| 1 | Sex (N = 1831) | | |
| | Male | 737 | 40.3 |
| | Female | 1094 | 59.7 |
| 2 | Level of Education (N = 1837) | | |
| | Junior High School | 984 | 53.6 |
| | Senior High School | 853 | 46.4 |
| 3 | Living Area (N = 1837) | | |
| | Urban | 719 | 39.1 |
| | Rural | 1118 | 60.9 |
| 4 | Level of mother's education (N = 1805) | | |
| | Low | 644 | 35.7 |
| | Intermediate | 844 | 46.8 |
| | High | 317 | 17.6 |
| 5 | Level of father's education (N = 1792) | | |
| | Low | 616 | 34.4 |
| | Intermediate | 765 | 42.7 |
| | High | 411 | 22.9 |

**Table 3. Distribution type of maltreatment among 12 to 14 years old (n = 984) and 15 to 17 years old (n = 853).**

| No. | Maltreatment | Response | | | | | |
|---|---|---|---|---|---|---|---|
| | | True and happened in last one year | | Kinda true and not happened in last one year | | Not True | |
| | | 12 to 14 years old N (%) | 15 to 17 years old N (%) | 12 to 14 years old N (%) | Senior N (%) | 12 to 14 years old N (%) | 15 to 17 years old N (%) |
| **Physical** | | | | | | | |
| 1. | Kicked you? | 317 (32.2%) | 183 (21.5%) | 56 (5.7%) | 47 (5.5%) | 610 (62%) | 622 (72.9%) |
| 2. | Shook you aggressively? | 292 (29.7%) | 166 (19.5%) | 49 (5%) | 36 (4.2%) | 642 (65.2%) | 650 (76.2%) |
| 3. | Slapped you on the face or on back of head? | 367 (37.3%) | 238 (27.9%) | 39 (4%) | 53 (6.2%) | 577 (58.6%) | 561 (65.8%) |
| 4. | Hit you on the head with knuckles? | 217 (22.1%) | 132 (15.5%) | 20 (2%) | 21 (2.5%) | 746 (75.8%) | 699 (81.9%) |
| 5. | Spanked you on the bottom with bare hand? | 219 (22.3%) | 135 (15.8%) | - | - | 764 (77.6%) | 717 (84.1%) |
| 6. | Hit you on the buttocks with an object (such as a stick, broom, cane, or belt)? | 161 (16.4%) | 91 (10.7%) | - | - | 822 (83.5%) | 761 (89.2%) |
| 7. | Hit you elsewhere (not buttocks) with an object (such as a stick, broom, cane, or belt)? | 86 (8.7%) | 28 (3.3%) | 13 (1.3%) | 6 (0.7%) | 884 (89.8%) | 818 (95.9%) |
| 8. | Hit you over and over again with object or fist ("beat-up")? | 16 (1.6%) | 17 (2%) | 3 (0.3%) | 3 (0.4%) | 964 (98%) | 832 (97.5%) |
| 9. | Choked you to prevent you from breathing? | 48 (4.9%) | 49 (5.7%) | 12 (1.2%) | 13 (1.5%) | 923 (93.8%) | 790 (92.6%) |
| 10. | Burned or scalded or branded you? | 64 (6.5%) | 76 (8.9%) | 12 (1.2%) | 15 (1.8%) | 907 (92.2%) | 761 (89.2%) |
| 11. | Put hot pepper, soap or spicy food in your mouth to cause you pain? | 499 (50.7%) | 327 (38.3%) | 90 (9.1%) | 117 (13.7%) | 394 (40%) | 408 (47.8%) |
| 12. | Locked you up or tied you to restrict movement? | 436 (44.3%) | 309 (36.2%) | 51 (5.2%) | 78 (9.1%) | 496 (50.4%) | 465 (54.5%) |
| 13. | Twisted your ear? | **657 (66.8%)** | **499 (58.5%)** | 74 (7.5%) | 94 (11%) | 252 (25.6%) | 259 (30.4%) |
| 14. | Pulled your hair? | 80 (8.1%) | 117 (13.7%) | 13 (1.3%) | 19 (2.2%) | 890 (90.4%) | 716 (83.9%) |
| 15. | Pinched you to cause pain? | 216 (22%) | 159 (18.6%) | 37 (3.8%) | 48 (5.6%) | 730 (74.2%) | 645 (75.6%) |
| 16. | Forced you to stand, sit or kneel in a position that caused pain? | 28 (2.8%) | 37 (4.3%) | 8 (0.8%) | 7 (0.8%) | 947 (96.2%) | 808 (94.7%) |
| 17. | Put you in time-out? | 23 (2.3%) | 30 (3.5%) | 6 (0.6%) | 9 (1.1%) | 954 (97%) | 813 (95.3%) |
| 18. | Withhold a meal as a punishment? | 336 (34.1%) | 189 (22.2%) | 42 (4.3%) | 54 (6.3%) | 605 (61.5%) | 609 (71.4%) |
| 19. | Give you drugs or alcohol? | 417 (42.4%) | 292 (34.2%) | 55 (5.6%) | 73 (8.6%) | 511 (51.9%) | 487 (57.1%) |
| **Psychological** | | | | | | | |
| 1. | Shouted, yelled, or screamed at you very loudly? | 434 (44.1%) | 373 (43.7%) | 121 (12.3%) | 118 (13.8%) | 429 (43.6%) | 362 (42.4%) |
| 2. | Insulted you by calling you dumb, lazy or other names like that? | 452 (45.9%) | 364 (42.7%) | 89 (9%) | 106 (12.4%) | 443 (45%) | 383 (44.9%) |
| 3. | Cursed you? | 256 (26%) | 152 (17.8%) | 50 (5.1%) | 52 (6.1%) | 678 (68.9%) | 649 (76.1%) |
| 4. | Ignored you? | 403 (41%) | 367 (43%) | 47 (4.8%) | 67 (7.9%) | 534 (54.3%) | 419 (49.1%) |
| 5. | Blamed you for his or her misfortune? | 120 (12.2%) | 73 (8.6%) | 24 (2.4%) | 26 (3%) | 840 (85.4%) | 754 (88.4%) |
| 6. | Forbade you from going out? | **574 (58.3%)** | **578 (67.8%)** | 69 (7%) | 50 (5.9%) | 341 (34.7%) | 225 (26.4%) |
| 7. | Embarrassed you publicly? | 335 (34%) | 272 (31.9%) | 57 (5.8%) | 72 (8.4%) | 592 (60.2%) | 272 (31.9%) |
| 8. | Said they wished you were dead or never been born? | 65 (6.6%) | 50 (5.9%) | 17 (1.7%) | 15 (1.8%) | 902 (91.7%) | 788 (92.4%) |
| 9. | Threatened to leave or abandon you? | 164 (16.7%) | 137 (16.1%) | 34 (3.5%) | 35 (4.1%) | 786 (79.9%) | 681 (79.8%) |
| 10. | Locked you out of the home? | 123 (12.5%) | 67 (7.9%) | 37 (3.8%) | 28 (3.3%) | 824 (83.7%) | 758 (88.9%) |
| 11. | Threatened to invoke harmful people, ghosts or evil spirits against you? | 75 (7.6%) | 64 (7.5%) | 20 (2%) | 35 (4.1%) | 889 (90.3%) | 754 (88.4%) |
| 12. | Threatened to hurt or kill you? | 58 (5.9%) | 53 (6.2%) | 16 (1.6%) | 12 (1.4%) | 910 (92.5%) | 788 (92.4%) |
| 13. | Referred to you skin color/gender/ religious or culture in a hurtful way | 177 (18%) | 87 (10.2%) | 32 (3.3%) | 26 (3%) | 775 (78.8%) | 740 (86.8%) |
| 14. | Tried to embarrass you because you were an orphan or without a parent? | 27 (2.7%) | 30 (3.5%) | 3 (0.3%) | 11 (1.3%) | 954 (97%) | 812 (95.2%) |

*(Continued)*

**Table 3.** (Continued)

| No. | Maltreatment | Response | | | | | |
|---|---|---|---|---|---|---|---|
| | | True and happened in last one year | | Kinda true and not happened in last one year | | Not True | |
| | | 12 to 14 years old N (%) | 15 to 17 years old N (%) | 12 to 14 years old N (%) | Senior N (%) | 12 to 14 years old N (%) | 15 to 17 years old N (%) |
| 15. | Stopped you from being with other children to make you feel bad or lonely? | 197 (20%) | 168 (19.7%) | 37 (3.8%) | 42 (4.9%) | 750 (76.2%) | 643 (75.4%) |
| 16. | Stole or broke or ruined your belonging? | 287 (29.2%) | 190 (22.3%) | 48 (4.9%) | 62 (7.3%) | 649 (66%) | 601 (70.5%) |
| 17. | Threatened you with bad marks that you didn't deserve? | 313 (31.8%) | 179 (21%) | 42 (4.3%) | 69 (8.1%) | 629 (63.9%) | 605 (70.9%) |

Table 4 shows the Rasch model analysis of maltreatment experienced among the first and second grades of junior school (range = 12–14) and the first and second grades of high school (range = 15 to 17).

Table 5 shows that 52.8% of adolescents aged 12 to 14 years and 59.2% of those aged 15 to 17 years exhibited abnormal emotions.

Table 6 shows significant correlation. However, the highest correlation value between psychological maltreatment with difficulty dimension is 0.148 with $p \leq 0.05$, obtained by conduct problem and indicating a weak correlation. The correlation value in senior high school is 0.047 with $p \leq 0.05$, indicating a weak correlation between physical maltreatment and the total problem. The highest correlation value between physical maltreatment with difficulty dimension is 0.048 with $p \leq 0.05$, obtained by conduct problem and indicating a weak correlation. The correlation value is 0.097 with $p \leq 0.05$, indicating a weak correlation between psychological maltreatment and the total problem. The highest correlation value between psychological maltreatment with difficulty dimensions is 0.069 with $p \leq 0.05$, obtained by conduct problem and indicating a weak correlation.

## Discussion

In this study, the prevalence of physical and psychological maltreatment among students aged 12 to 14 years old and 15 to 17 years old was higher than other studies in Tanzania (76.6% of 1000 respondents) and Iran (<50% of 738 respondents) [26–30]. Meanwhile, a KPAI survey reported that the prevalence of maltreatment was 91%, assuming that most children had experienced maltreatment. Moreover, the prevalence of mental health problems in junior and senior high schools in this study was more than fifth percent, as was the case in Oman, India, and China, but was higher than that in studies in Austria and Mongolia [13, 31–36]. The high prevalence of physical maltreatment can be attributed to parents' or guardians' knowledge of the definition of maltreatment. Parents may consider this as a culture to discipline their children.

Physical maltreatment may affect mental health, causing anxiety, depression, and low self-esteem, as well as poor mental health in adolescents [37]. Childhood physical maltreatment

**Table 4. Experience of maltreatment based on Rasch model analysis among 12 to 14 years old (n = 984) and 15 to 17 years old (n = 853).**

| No | Maltreatment | No | | yes | |
|---|---|---|---|---|---|
| | | 12 to 14 years old | 15 to 17 years old | 12 to 14 years old | 15 to 17 years old |
| | | N (%) | N (%) | N (%) | N (%) |
| 1 | Physical | 142 (14.4%) | 145 (17%) | **842 (85.6%)** | **708 (83%)** |
| 2 | Psychological | 104 (10.6%) | 146 (17.1%) | **880 (89.4%)** | **707 (82.9%)** |

**Table 5. Mental health problems among 12 to 14 years old (n = 984) and 15 to 17 years old (n = 853).**

| No | Mental Health Problems | No | | Yes | |
|---|---|---|---|---|---|
| | | 12 to 14 years old | 15 to 17 years old | 12 to 14 years old | 15 to 17 years old |
| | | N (%) | N (%) | N (%) | N (%) |
| 1 | Emotional | 715 (72.7%) | 595 (69.8%) | 269 (27.3%) | 258 (30.2%) |
| 2 | Conduct | 678 (68.9%) | 657 (77%) | 306 (31.1%) | 196 (23%) |
| 3 | Hyperactivity | 595 (60.4%) | 429 (50.3%) | 389 (39.6%) | 424 (49.7%) |
| 4 | Peer Relationship | 328 (33.3%) | 226 (26.5%) | 656 (66.7%) | 627 (73.5%) |
| 5 | Total Problems | 464 (47.2%) | 348 (40.8%) | **520 (52.8%)** | **505 (59.2%)** |
| 6 | Prosocial | 837 (85.1%) | 786 (92.1%) | 147 (14.9%) | 67 (7.9%) |

may increase the risk of developing psychiatric disorders such as anxiety, depression, and post-traumatic stress disorder in adulthood [38–40]. Our results showed a significant correlation between psychological maltreatment and mental health problems. A previous study also showed a significant correlation between psychological maltreatment with internalizing problems (withdrawal, anxiety or depression) and somatic symptom and externalizing (aggressive symptoms and criminal behavior) [41–51].

Physical and psychological maltreatment were significantly correlated with conduct problems. Docherty et al. indicated that children who experienced maltreatment in early childhood were more likely to experience conduct problems and decreased guilt during adolescence. Adolescents who exhibited conduct problems and appeared to lack guilt were more likely to have a personal history of maltreatment [52]. Parents with mental health problems, such as depression, social isolation, and drug use, have a significant effect on their child's health. Parents with a history of mental illness tend to behave negatively towards their children, such as being sensitive to their children's behavior and being inconsistent in educating their children. As a result, the appearance of emotional problems and changes in children's behavior as well as genetic transmission and stressors given by parents to their children [52, 53]. Several studies support this statement by highlighting the significant correlation between mental health problems and the emergence of psychological abuse and neglect.

**Table 6. Correlation between physical and psychological maltreatment with mental health problems among 12 to 14 years old (n = 984) and 15 to 17 years old (n = 853).**

| No | Mal- treatment type | Mental Health Problems | Correlation Value | | | | | | | |
|---|---|---|---|---|---|---|---|---|---|---|
| | | | Regression Coef. | | Correlation Coef. | | Adj. $R^2$ | | CI = 95% | |
| | | | 12 to 14 years old | 15 to 17 years old | 12 to 14 years old | 15 to 17 years old | 12 to 14 years old | 15 to 17 years old | 12 to 14 years old | 15 to 17 years old |
| 1 | Physical | Emotional | 0.338 | 0.168 | 0.299 | 0.134 | 0.089 | 0.017 | 0.270–0.405 | 0.084–0252 |
| | | Conduct | 0.309 | 0.237 | 0.028 | 0.271 | **0.109** | **0.048** | 0.254–0.364 | 0.167–0.308 |
| | | Hyperactivity | 0.176 | 0.086 | 0.199 | 0.098 | 0.039 | 0.008 | 0.122–0.230 | 0.027–0.144 |
| | | Peer Relationship | 0.244 | 0.123 | 0.043 | 0.148 | 0.020 | 0.021 | 0.144–0.281 | 0.068–0.179 |
| | | Total Problems | 0.224 | 0.120 | 0.386 | 0.220 | **0.148** | **0.047** | 0.190–0.257 | 0.084–0.156 |
| 2 | Psychological | Emotional | 0.483 | 0.340 | 0.351 | 0.222 | 0.123 | 0.048 | 0.402–0.564 | 0.239–0.441 |
| | | Conduct | 0.438 | 0.347 | 0.386 | 0.265 | **0.148** | **0.069** | 0.372–0.503 | 0.262–0.432 |
| | | Hyperactivity | 0.254 | 0.173 | 0.236 | 0.163 | 0.055 | 0.025 | 0.189–0.319 | 0.103–0.244 |
| | | Peer Relationship | 0.318 | 0.207 | 0.297 | 0.204 | 0.087 | 0.040 | 0.254–0.382 | 0.140–0.274 |
| | | Total Problems | 0.310 | 0.209 | 0. 439 | 0.312 | **0.191** | **0.097** | 0.270–0.350 | 0.167–0.252 |

*Linear Regression Correlation Test p≤0.05, all significant

This study provides information to increase the awareness of the government and society about rampant maltreatment accompanied by mental health disorders experienced by adolescents. The results of this study can be used as basic data and initial intervention steps by the government to develop program-based policies intended for families, school institutions, and communities. The policies can aim to decrease maltreatment and mental health disorders in adolescents through good relationships between parents and children or by conducting positive discipline for children.

The strength of this study is in the sample size. In addition, the separation of categories in this study as junior and high school levels was analyzed in anticipation of bias resulting from an adolescent developmental phase. Since the three psychosocial developmental phases of adolescence are the early phase or middle school (10–14 years age), middle phase or high school (15–18 years age), and late phase approximating college years, a trajectory between phases involves changes in cognitive, emotional, and social behaviors [53].

Participation and information biases may be considered the limitations of this study. Another limitation of the study is that the correlation between maltreatment and mental health problems is weak. This study has not considered other risk factors, such as age, parent's occupation, socioeconomic factors, and family characteristics of respondents, which may contribute to the low correlation strength observed. Furthermore, this study did not identify the perpetrators of maltreatment. The high incidence of maltreatment in this study was not accompanied by the frequency of maltreatment; therefore, the frequency of maltreatment should be considered in future studies.

## Conclusions

Physical and psychological maltreatment are correlated with mental health problems in adolescents. Adolescents with a history of physical and psychological maltreatment had a higher correlation with behavioral and emotional problems. Further research should include other risk factors in its analysis.

## Supporting information

**S1 Table. Validity and reliability of the instruments.**
(DOCX)

## Author Contributions

**Conceptualization:** Nita Arisanti, Meita Dhamayanti.

**Data curation:** Nita Arisanti, Anisa Swediana, Deborah Karubaba, Anindita Noviandhari, Meita Dhamayanti.

**Formal analysis:** Nita Arisanti, Deni K. Sunjaya, Meita Dhamayanti.

**Funding acquisition:** Meita Dhamayanti.

**Investigation:** Anisa Swediana, Deborah Karubaba, Meita Dhamayanti.

**Methodology:** Nita Arisanti, Deni K. Sunjaya, Meita Dhamayanti.

**Project administration:** Nita Arisanti, Anisa Swediana, Deborah Karubaba, Anindita Noviandhari, Meita Dhamayanti.

**Resources:** Meita Dhamayanti.

**Software:** Deni K. Sunjaya, Meita Dhamayanti.

**Supervision:** Nita Arisanti, Anisa Swediana, Deborah Karubaba, Anindita Noviandhari, Meita Dhamayanti.

**Validation:** Nita Arisanti, Deborah Karubaba, Anindita Noviandhari, Deni K. Sunjaya, Meita Dhamayanti.

**Visualization:** Meita Dhamayanti.

**Writing – original draft:** Nita Arisanti, Deni K. Sunjaya, Meita Dhamayanti.

**Writing – review & editing:** Nita Arisanti, Anisa Swediana, Deborah Karubaba, Anindita Noviandhari, Meita Dhamayanti.

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
