## [Decision Letter · Decision Letter 0]

19 Jan 2024

PONE-D-23-24880Relationship between maltreatment and mental health in adolescents: a school-based study from IndonesiaPLOS ONE

Dear Dr. Dhamayanti,

Thank you for submitting your manuscript to PLOS ONE. After careful consideration, we feel that it has merit but does not fully meet PLOS ONE’s publication criteria as it currently stands. Therefore, we invite you to submit a revised version of the manuscript that addresses the points raised during the review process.

We look forward to receiving your revised manuscript.

Kind regards,

Gilbert Sterling Octavius

Academic Editor

PLOS ONE

Journal Requirements:

"The study was supported financially by an Academic Leadership Grant from Universitas Padjadjaran undernamed Meita Dhamayanti,  and the APC was funded by Academic Leadership Grant from Universitas Padjadjaran

1 9 5 9 /UN6.3.1/PT.00/2021"

Reviewers' comments:

Reviewer's Responses to Questions

**Comments to the Author**

1. Is the manuscript technically sound, and do the data support the conclusions?

Reviewer #1: Partly

Reviewer #2: Partly

2. Has the statistical analysis been performed appropriately and rigorously? 

Reviewer #1: No

Reviewer #2: Yes

3. Have the authors made all data underlying the findings in their manuscript fully available?

Reviewer #1: Yes

Reviewer #2: Yes

4. Is the manuscript presented in an intelligible fashion and written in standard English?

Reviewer #1: Yes

Reviewer #2: No

5. Review Comments to the Author

Reviewer #1: This is potentially interesting, however, the manuscript as presented has some problems that need to be addressed before it is publishable, those problems are ones of context and analyses.

With respect to context as a reader I would like more details as the nature of the schools sampled in this study.

Context – The school system – what types of schools were sampled? Government, private, religious? Was the curriculum in the schools the nation one form the Ministry of Education? where any “national plus”? While recognizing that education is compulsory (to age 12?) what is the participation rate for the age groups sampled in this study? Basically, how generalizable is the sample?

Analyses – With respect to data analysis I am surprised that there is no stratified analysis of the data by gender? The literature generally shows that boys and girls experience different types and levels of abuse.

I was also surprised that sexual abuse/maltreatment, a major of abuse/maltreatment was not reported on. Was data on sexual abuse not collected? Why?

The analyses reported on in the manuscript was largely descriptive – frequency of responses to specific items in the ICAST-C. A more detailed analysis is warranted. What of correlates of abuse/maltreatment i.e., students’ marks, parents’ education, income, type of school etc. ?

The written English of the manuscript is generally good but there is occasionally odd phrasing – “…was expert panel translated…” should read ‘…was translated by an expert panel…’ The comment on about the high population of adolescents in West Java on the top of the page 4 line 69 should be deleted.

Reviewer #2: This is a large representative study investigating the prevalence of physical and psychological abuse and their association with mental health problems among Indonesian teenagers. The findings provided evidence of association between both types of abuse and different types of psychological difficulties (i.e, conduct disorders symptoms, hyperactivity symptoms, emotional symptoms, and peer relationship difficulties). The sample is sizeable, the methods robust, and the findings contribute to informing about the prevalence and the impact of childhood maltreatment in low and middle income countries.

1.The Introduction might be improved by including references to large epidemiological studies and systematic reviews on the association between childhood maltreatment and mental health outcomes. It would be useful to state clearly what this study adds to the existing literature on childhood maltreatment.

2.I suggest adding at the end of the introduction specific hypotheses regarding the relationship between specific types of childhood maltreatment and mental health problems.

3.The Measures should present information about the sociodemographic variables (e.g., definition of “low” parental education level and ‘rural rea’). I would suggest incorporating the psychometric properties of the ICAST and the SDQ in this section, rather than in the results.

4.If possible, please provide information about possible participants bias, i. e., differences between participants and those who refused to participate in the study.

5.The prevalence of physical and psychological abuse is quite high. I would suggest commenting these results in the discussion. I also wonder if cut-offs for severe abuse in the ICAST scale have been defined in literature. If possible, I suggest repeating the analyses with a stricter definition of childhood abuse.

6.The role of potential confounders is poorly investigated. I suggest testing the association of childhood abuse and mental health problems with sex, age, urbanicity, and parental education level (a proxy of social class). In addition to the crude association between childhood abuse and mental health problems, would be important to report multivariated analyses adjusted for socio demographic variable.

7.Please, include in the limitations a brief discussion about participation bias, information bias, and cross-sectional study design.

8.The manuscript is not always clear. A careful revision would address some linguistic choices (e.g., ‘mental-emotional states’) and minor grammar mistakes (e.g., ‘A previous study also showed a significant correlation…’ referred to multiple studies).

6. PLOS authors have the option to publish the peer review history of their article (what does this mean?). If published, this will include your full peer review and any attached files.

Reviewer #1: No

Reviewer #2: No

---

## [Author Response · Author response to Decision Letter 0]

30 May 2024

Response Letter

Dear Editor and Reviewer,

I am very much thankful to the editor and reviewer for thorough review. I have revised my present manuscript. I hope my revision has improved the manuscript to a level of their satisfaction. 

Reviewer reports:

Reviewer #1: This is potentially interesting, however, the manuscript as presented has some problems that need to be addressed before it is publishable, those problems are ones of context and analyses.

With respect to context as a reader I would like more details as the nature of the schools sampled in this study.

Context – The school system – what types of schools were sampled? Government, private, religious? Was the curriculum in the schools the nation one form the Ministry of Education? where any “national plus”? While recognizing that education is compulsory (to age 12?) what is the participation rate for the age groups sampled in this study? Basically, how generalizable is the sample?

Response:

The authors thank to reviewer for the comments. 

What types of schools were sampled? Government, private, religious? The government and private schools was chosen in this study. (page 3, line 67)

Was the curriculum in the schools the nation one form the Ministry of Education? where any “national plus”? Yes, the schools implemented national curriculum from the Ministry of Health. 

While recognizing that education is compulsory (to age 12?) what is the participation rate for the age groups sampled in this study? The participation rate was 53 % (1837 of 3452 filled the questionaries). (page 6, line 124)

Analyses – With respect to data analysis I am surprised that there is no stratified analysis of the data by gender? The literature generally shows that boys and girls experience different types and levels of abuse.

I was also surprised that sexual abuse/maltreatment, a major of abuse/maltreatment was not reported on. Was data on sexual abuse not collected? Why?

Response:

The authors thank to reviewer for the comments. We did not show the stratified analysis of the data by gender because the result of DIF analysis from ICAST-C questionnaires was not significant. 

The data on sexual abuse was collected in this study but in this manuscript, we focus on physical and psychological maltreatment. 

The written English of the manuscript is generally good but there is occasionally odd phrasing – “…was expert panel translated…” should read ‘…was translated by an expert panel…’ The comment on about the high population of adolescents in West Java on the top of the page 4 line 69 should be deleted.

Response:

The authors thank to reviewer for the comments. It has been revised and deleted.

Reviewer #2: This is a large representative study investigating the prevalence of physical and psychological abuse and their association with mental health problems among Indonesian teenagers. The findings provided evidence of association between both types of abuse and different types of psychological difficulties (i.e, conduct disorders symptoms, hyperactivity symptoms, emotional symptoms, and peer relationship difficulties). The sample is sizeable, the methods robust, and the findings contribute to informing about the prevalence and the impact of childhood maltreatment in low and middle income countries.

1.The Introduction might be improved by including references to large epidemiological studies and systematic reviews on the association between childhood maltreatment and mental health outcomes. It would be useful to state clearly what this study adds to the existing literature on childhood maltreatment.

Response:

The authors thank to reviewer for the comments. It has been added with new references about the association between childhood maltreatment and mental health outcomes. (page 3, line 55).

The authors used SDQ questionnaire to assess the categories of mental health problems and we believe that the study findings will add to the existing literature. (page 3, line 61-62).

2.I suggest adding at the end of the introduction specific hypotheses regarding the relationship between specific types of childhood maltreatment and mental health problems.

Response:

The authors thank to reviewer for the comments. It has been added in page 3, line 55-56.

3.The Measures should present information about the sociodemographic variables (e.g., definition of “low” parental education level and ‘rural rea’). I would suggest incorporating the psychometric properties of the ICAST and the SDQ in this section, rather than in the results.

Response:

The authors thank to reviewer for the comments. It has been replaced to methods section. 

4.The prevalence of physical and psychological abuse is quite high. I would suggest commenting these results in the discussion. I also wonder if cut-offs for severe abuse in the ICAST scale have been defined in literature. If possible, I suggest repeating the analyses with a stricter definition of childhood abuse.

Response:

The authors thank to reviewer for the comments. The discussion has been added in the discussion (page 8, line 159).

There is no cut off to describe the definition of child abuse based the ICAST questionnaire and this is the reason why the study used Rasch model analysis. (page 5, line 98-99)

5.Please, include in the limitations a brief discussion about participation bias, information bias, and cross-sectional study design.

Response:

The authors thank to reviewer for the comments. It has been added in the limitation part. (page 9, line 191)

6.The manuscript is not always clear. A careful revision would address some linguistic choices (e.g., ‘mental-emotional states’) and minor grammar mistakes (e.g., ‘A previous study also showed a significant correlation…’ referred to multiple studies).

Response:

The authors thank to reviewer for the comments. The grammatical errors have been corrected,

I hope the revised version is now suitable for publication and look forward to hearing from you in due course.

Sincerely,

Meita Dhamayanti

Corresponding author

---

## [Decision Letter · Decision Letter 1]

28 Jun 2024

PONE-D-23-24880R1Relationship between Maltreatment and Mental Health in Adolescent: A School Based Study from IndonesiaPLOS ONE

Dear Dr. Dhamayanti,

Thank you for submitting your manuscript to PLOS ONE. After careful consideration, we feel that it has merit but does not fully meet PLOS ONE’s publication criteria as it currently stands. Therefore, we invite you to submit a revised version of the manuscript that addresses the points raised during the review process.

We look forward to receiving your revised manuscript.

Kind regards,

Gilbert Sterling Octavius

Academic Editor

PLOS ONE

Journal Requirements:

Reviewers' comments:

Reviewer's Responses to Questions

**Comments to the Author**

1. If the authors have adequately addressed your comments raised in a previous round of review and you feel that this manuscript is now acceptable for publication, you may indicate that here to bypass the “Comments to the Author” section, enter your conflict of interest statement in the “Confidential to Editor” section, and submit your "Accept" recommendation.

Reviewer #2: (No Response)

2. Is the manuscript technically sound, and do the data support the conclusions?

Reviewer #2: (No Response)

3. Has the statistical analysis been performed appropriately and rigorously? 

Reviewer #2: (No Response)

4. Have the authors made all data underlying the findings in their manuscript fully available?

Reviewer #2: (No Response)

5. Is the manuscript presented in an intelligible fashion and written in standard English?

Reviewer #2: (No Response)

6. Review Comments to the Author

Reviewer #2: Thank you for carrying out some of the suggested changes. However, I could not find the point where the authors state the study hypotheses. Moreover the discussion about participation and information bias (as well as the study design) should be corroborated with at least one more sentence briefly explaining in which way the results of this study might be affected by these bias. Once addressed these points I am happy to endorse the submission

7. PLOS authors have the option to publish the peer review history of their article (what does this mean?). If published, this will include your full peer review and any attached files.

Reviewer #2: No

---

## [Author Response · Author response to Decision Letter 1]

24 Jul 2024

Reviewer #2: 

Point 1: Thank you for carrying out some of the suggested changes. However, I could not find the point where the authors state the study hypotheses. 

Response 1: Thank you very much for your comment. We carefully considered your suggestion and added the hypothesis in INTRODUCTION, line 75

This study hypothesis that physical and psychological maltreatment correlates with mental health problem in adolescent.

Point 2: Moreover the discussion about participation and information bias (as well as the study design) should be corroborated with at least one more sentence briefly explaining in which way the results of this study might be affected by these bias. Once addressed these points I am happy to endorse the submission 

Response 2: Thank you very much for your comment. 

the results of this study might be affected by these bias, the statement has also added in DISSCUSION (line 205 -210) as follow:

Also, the separation of categories in this study as the junior and high school level is analyzed in anticipation of bias resulted from an adolescence developmental phase. Since three psychosocial developmental phase of adolescence are early phase or middle school (10-14 years age) , middle phase or high school (15-18 years age) and late phase approximate college years, hence a trajectory between phase involve amount of change in cognitive, emotional, and social behavior [53]. The participation and information bias may lead to the limitation of this study.

---

## [Editor Report · Decision Letter 2]

7 Aug 2024

PONE-D-23-24880R2Relationship between Maltreatment and Mental Health in Adolescent: A School Based Study from IndonesiaPLOS ONE

Dear Dr. Dhamayanti,

Thank you for submitting your manuscript to PLOS ONE. After careful consideration, we feel that it has satisfied our scientific requirements for publication.

However, our editorial team have significant concerns about the grammar, usage, and overall readability of the manuscript. PLOS ONE requires that published manuscripts use language which is 'clear, correct, and unambiguous', see our criteria for publication at https://journals.plos.org/plosone/s/criteria-for-publication#loc-5. We therefore request that you revise the text to fix the grammatical errors and improve the overall readability of the text.

We suggest you have a fluent English-language speaker thoroughly copyedit your manuscript for language usage, spelling, and grammar. If you do not know anyone who can do this, you may wish to consider employing a professional scientific editing service.

Whilst you may use any professional scientific editing service of your choice, PLOS has partnered with both American Journal Experts (AJE) and Editage to provide discounted services to PLOS authors. Both organizations have experience helping authors meet PLOS guidelines and can provide language editing, translation, manuscript formatting, and figure formatting to ensure your manuscript meets our submission guidelines. To take advantage of our partnership with AJE, visit the AJE website (https://www.aje.com/go/plos/) for a 15% discount off AJE services. To take advantage of our partnership with Editage, visit the Editage website (www.editage.com) and enter referral code PLOSEDIT for a 15% discount off Editage services. If the PLOS editorial team finds any language issues in text that either AJE or Editage has edited, the service provider will re-edit the text for free.

Please note that we will not be able to proceed with publication of your manuscript until the concerns above are addressed.

* A copy of your manuscript showing your changes by either highlighting them or using track changes (uploaded as a supporting information file)

* A clean copy of the edited manuscript (uploaded as the new manuscript file)

We look forward to receiving your revised manuscript.

Kind regards,

Miquel Vall-llosera Camps

Senior Staff Editor

PLOS ONE

on behalf of

Gilbert Sterling Octavius

Academic Editor

PLOS ONE 
---

## [Author Response · Author response to Decision Letter 2]

27 Aug 2024

Upon this resubmission , we provide:

* The EDITAGE professional service that edited our manuscript

* A copy of manuscript with tract changes by either highlighting them or using track changes (uploaded as a supporting information file)

* A clean copy of the edited manuscript (uploaded as the new manuscript file)

---

## [Editor Report · Decision Letter 3]

3 Sep 2024

Relationship between maltreatment and mental health in adolescents: a school-based study in Indonesia

PONE-D-23-24880R3

Dear Dr. Dhamayanti,

We’re pleased to inform you that your manuscript has been judged scientifically suitable for publication and will be formally accepted for publication once it meets all outstanding technical requirements.

Kind regards,

Gilbert Sterling Octavius

Academic Editor

PLOS ONE
---

## [Editor Report · Acceptance letter]

10 Sep 2024

PONE-D-23-24880R3 

PLOS ONE

Dear Dr. Dhamayanti, 

I'm pleased to inform you that your manuscript has been deemed suitable for publication in PLOS ONE. Congratulations! Your manuscript is now being handed over to our production team.

Kind regards, 

on behalf of

Dr. Gilbert Sterling Octavius 

Academic Editor

PLOS ONE